# Levels of Circulating IgM and IgY Natural Antibodies in Broiler Chicks: Association with Genotype and Farming Systems

**DOI:** 10.3390/biology12020304

**Published:** 2023-02-14

**Authors:** Ioannis Sarrigeorgiou, Theodora Stivarou, Gerasimina Tsinti, Apostolos Patsias, Evgenia Fotou, Vasiliki Moulasioti, Dimitra Kyriakou, Constantinos Tellis, Maria Papadami, Vassilios Moussis, Vasileios Tsiouris, Vassilios Tsikaris, Demokritos Tsoukatos, Peggy Lymberi

**Affiliations:** 1Immunology Laboratory, Immunology Department, Hellenic Pasteur Institute (HPI), 127, Vasilissis Sofias Avenue, 11521 Athens, Greece; 2Microbiology and Chemical Laboratory, Pindos APSI, 45500 Rodotopi Ioannina, Greece; 3Department of Chemistry, Section of Organic Chemistry and Biochemistry, University of Ioannina, 45110 Ioannina, Greece; 4Unit of Avian Medicine, Faculty of Veterinary Medicine, School of Health Sciences, Aristotle University of Thessaloniki, 54124 Thessaloniki, Greece

**Keywords:** natural antibodies, IgM and IgY antibodies, innate immunity, broilers, poultry industry, alternative housing systems, free range, domestic plant extracts, diet supplements, oregano oil

## Abstract

**Simple Summary:**

Poultry is currently the most efficient animal productive system and could contribute to fulfilling the need for protein in order to supply the increasing human population. However, intensive selection of broilers makes them more susceptible to several welfare concerns. Moreover, in order to improve animal welfare, alternative housing systems, such as free-range and organic systems, are becoming increasingly popular. These changes in the modern poultry industry, in combination with restrictions on antibiotics, have led to the need for poultry with higher disease resistance. Naturally occurring antibodies (NAbs), which are major components of innate immunity, have been previously detected in chicken sera. These studies showed that NAbs are heritable and that high levels of NAbs are associated with survival and highly specific antibody responses against pathogens. Therefore, we investigated the circulating levels of IgM and IgY NAbs against selected antigens (actin, DNA, trinitrophenol and lipopolysaccharide) as potential biomarkers of poultry welfare and productivity in fast-growth (Ross 308) and slow-growth (Sasso) broilers raised in conventional and free-range systems, respectively, under 3-year industrial-scale production. Overall, we demonstrated significant differences in IgM NAb levels during the final breeding step between the two commercial genotypes, highlighting NAbs as potential biomarkers to be exploited in the poultry industry.

**Abstract:**

Naturally occurring antibodies (NAbs), which are major components of innate immunity, exist in circulation under healthy conditions without prior antigenic stimulation and are able to recognize both self- and non-self-constituents. The present study aimed at identifying potential immunological differences between commercial fast- and slow-growth broilers (*n* = 555) raised in conventional and free-range systems, respectively, through the use of the specificity, isotypes and levels of circulating NAbs. The possible beneficial effect of oregano-based dietary supplementation was also evaluated. To this end, serum IgM and IgY NAbs against self- (actin and DNA) and non-self- antigens (trinitrophenol and lipopolysaccharide) were measured by ELISA and further correlated with genotype, season and performance. Significantly higher levels of IgM NAbs against all antigens were found in slow-growth compared to fast-growth broilers. IgM NAb levels were also significantly increased in dietarily supplemented slow-growth broilers versus those consuming standard feed. Moreover, significantly elevated levels of anti-DNA IgY NAbs were found in fast-growth compared to slow-growth broilers, whereas the opposite was observed for anti-LPS IgY NAbs. Multivariate linear regression analysis confirmed multiple interactions between NAb levels, genotype, season and performance. Overall, serum NAbs have proven to be valuable innovative immunotools in the poultry industry, efficiently differentiating fast-growing versus slow-growing broilers, and dietary supplementation of plant extracts can enhance natural immunity.

## 1. Introduction

Poultry is currently the most rapidly growing animal productive system and could contribute to fulfilling the need for protein in order to supply the steadily increasing human population. Since the mid-twentieth century, an intensive selection production process has been carried out, resulting in more efficient chickens in terms of feed conversion and rapid growth, making them an effective system for high-quality protein production. However, this intensive selection of broilers has made them more susceptible to several welfare concerns [1]. Under commercial housing conditions, in order to improve animal welfare, alternative housing systems are becoming increasingly popular [2]. In general, birds kept in alternative systems can express their full natural behavioral repertoire, displaying more comfort and activity than in conventional systems. On the other hand, birds kept in alternative systems show more aggression, feather pecking and cannibalism, as well as a higher incidence of diseases [3].

Changes in the modern poultry industry, in combination with the recent developments in restricted preventive medicine, have led to the need for poultry with higher disease resistance [1,2]. Therefore, a fully functional immune system is essential, which may be threatened by stressors and other adverse environmental stimuli, thereby affecting vaccination response, disease susceptibility and survival [4]. Additionally, efforts to increase performance in broiler chicks often involve dietary supplements based on domestic plant extracts that potentially enhance birds’ responsiveness under stressful environmental stimuli.

Naturally occurring antibodies (NAbs), which are important players in both innate and acquired immunity, belong to M, G, A and E immunoglobulin classes and are defined as germline encoded antibodies found in every vertebrate species tested, existing in circulation under healthy conditions without prior known exogenous antigenic stimulation [5]. NAbs are polyreactive and are able to bind self- and non-self-components, such as nucleic acids, proteins and haptens, and act as a first-line immune defense against pathogens, with a major role in clearance and immune homeostasis [5,6,7,8]. The presence of IgM and IgY NAbs in chicken sera has been previously demonstrated using a series of target antigens, with keyhole limpet hemocyanin (KLH) being the most studied [9,10,11,12,13]. KLH is regularly used as an example of a “naïve” antigen for detection of NAbs in poultry, as chickens are not normally exposed to it [14]. Interestingly, levels of NAbs binding to KLH in chickens were previously shown to be heritable and to be associated with survival and disease resistance, with highly specific antibody responses against pathogens [8,15,16,17]. These data strongly suggest that the levels and specificity of Nabs are directly related to immune system efficiency and, therefore, poultry welfare and productivity. Additionally, regarding animal robustness, studies report that the use of plant extracts as feed additives may also have a positive effect on birds’ health, as well as pathogen protection, with a positive effect on productivity [18,19].

Therefore, we investigated circulating IgM and IgY NAb levels as potential biomarkers of poultry welfare and productivity in two commercial genotypes (Ross 308 and Sasso) raised in either conventional or alternative (free-range) systems under 3-year industrial-scale production. The effect of domestic plant extracts (oregano derivatives) as diet supplements on the immune status of free-range poultry was also studied. For the evaluation of NAb specificity, we focused on the IgM and IgY NAbs, as IgM is the major immunoglobulin class of NAbs in serum, and it biological roles have been mainly attributed to its function and characteristics, such as polyreactivity. IgY NAbs also occur in vivo and can serve as an index of an organism’s immunological history. For the evaluation of NAb levels, common and conserved self-antigens such as actin and native DNA, as well as non-self-antigens such as lipopolysaccharide (LPS) and trinitrophenol (TNP), were chosen. Cytoskeletal proteins (e.g., actin and myosin) are frequent targets of NAbs and are thought to perform homeostatic roles within the immune response [20], while DNA represents a ‘self-antigen’ often released into the extracellular milieu from necrotic cells and therefore serves as an indirect measure of stress and cytotoxicity [14]. On the other hand, LPS is a common pathogen-associated molecular pattern existing in Gram-negative bacteria and strongly interacting with innate cell receptors (e.g., TLRs) and is therefore used as a measure of immune alertness and activation [11]. Furthermore, TNP, an organic molecule to which living organisms are not normally exposed, has been included as a major target of NAbs and as a measure of serum polyreactivity [21,22].

Therefore, with the proposed protocol, we aimed to identify potential differences in the levels and specificity of NAbs by using informative antigenic targets of self- and non-self-constituents that could be further utilized to discriminate fast-growing Ross 308 chicks raised in conventional systems from slow-growing Sasso chicks raised in free-range systems during the final stage of farm breeding.

## 2. Materials and Methods

### 2.1. Study Design

This study was set up to investigate three study groups under industrial-scale meat production (Pindos APSI, Ioannina, Greece): fast-growth broilers (Ross 308, Aviagen Group, Huntsville, AL, USA) raised in conventional systems constituted the C group, slow-growth broilers (Sasso, Hendrix Genetics BV, Boxmeer, The Netherlands) raised in free-range systems constituted the FR group and slow-growth broilers (Sasso, Hendrix Genetics BV, The Netherland) raised in free-range systems with 5% domestic oregano derivatives incorporated into their feed constituted the FRp group. Oregano derivatives (oregano oil) were purchased by local farmers in the agricultural area of Ioannina, Epirus and used as premixed (with food) dietary supplements. Broilers of different genotypes were allocated to different facilities (stocking density: C group, 15 birds/m^2^; FR and FRp groups, 13 birds/m^2^ indoors and 1 bird/m^2^ in forage paddock) and raised under the same conditions (ventilation, lighting, drinking water, etc.) with free access to feed and water. Birds were placed in commercial poultry farms on the same date, which were fully equipped with automatic ventilation, heating, lighting and feeding systems. Water and feed were offered to all birds ad libitum, and the lighting program and microenvironmental conditions (temperature, humidity, CO_2_ and NH_3_) were automatically regulated in all houses according to the recommendations of the breeding company [23]. Commercial diets were designed for each group (genotype), age and period according to the nutritional specifications shown in Table 1. Diets for all groups were wheat-and maize-based formulations. The composition and chemical analysis of the three different animal feeds are presented in Appendix A. Slow-growth chickens were released (by will) for outdoor pasture from the 28th day of age until their slaughter. As the slaughter age dependes on market needs and consumer preferences, the most common commercial final body weight in Greece is 2.5–3.0 Kg, which is reached at about 45 days of age for fast-growth broilers (Ross 308) and approximately 65 days of age for slow-growth broilers (Sasso).

This was a cohort study design consisting of 14 breeding periods of industrial scale between 2018 and 2021 (Appendix A). During this period, serum samples from 210 C, 210 FR and 135 FRp chickens were collected at 47 and 67 days of age for fast- and slow-growth broilers, respectively, before animals were moved to the slaughterhouse. Blood samples were gathered by jugular vein puncture, and tubes were centrifuged at 1500 rpm for 10 min at 4 °C. Supernatants (serum) were stored at −80 °C until use.

The mortality of each group was recorded daily, and the body weight (BW), feed conversion ratio (FCR) and European Production Efficiency Factor (EPEF) were calculated cumulatively at the end of the farm breeding. Feed intake (FI) and body weight gain (BWG) were monitored per growing period. FCR and EPEF were calculated cumulatively for the whole breeding period according to the Equations (1) and (2). Final values of performance parameters are shown in Table 2.
(1)$$FCR=\frac{FI}{BWG}$$
(2)$$EPEF=\frac{BW\;\left(kg\right)\times Liveability\;\left(\%\right)}{FCR\times slaughter\;age\;\left(d\right)}\times 100$$

### 2.2. Development of In-House ELISAs

Actin (A3653) from bovine muscle, deoxyribonucleic acid sodium salt (native DNA) from calf thymus (D1501) and lipopolysaccharide (LPS) from *Escherichia coli* (L2880) were purchased from Sigma Aldrich (St. Louis, MO, USA). The hapten trinitrophenol (TNP) was used in ELISA coupled to a carrier molecule, namely bovine serum albumin (BSA), and TNP_25_-BSA conjugate was prepared as previously described [21]. High binding ELISA microplates (Nunc-MaxiSorp, Roskilde, Denmark) were used for antigen coating. Briefly, 100 μL/well of antigens diluted at 10 μg/mL final concentration in carbonate-bicarbonate buffer 0.1 M pH 9.6 was incubated overnight at 4 °C in ELISA plates. Plates were then thoroughly washed with phosphate-buffered saline (PBS, pH 7.4) and saturated with 150 μL/well PBS containing 1% BSA for 1 h at 37 °C. Sera were diluted 1/100 in sample dilution buffer (blocking buffer containing 0.05% Tween), and 100 μL/well was incubated for 2 h at 37 °C. After extensive wash, alkaline phosphatase conjugated secondary antibodies against chicken IgY (303-055-008, Jackson ImmunoResearch, West Grove, Pennsylvania, USA) or IgM (SAB3700239-1, Sigma Aldrich, St. Louis, MO, USA) were added for 0.1 and 0.5 μg/mL final concentrations, respectively (100 μL/well) and incubated for 2 h at 37 °C. After extensive washing, antibody binding was assessed with the addition of substrate 4-nitrophenyl-phosphate-disodium salt hexahydrate (pNPP-N2765, Sigma), and the optical density (OD) of colored product was measured at 450 nm (620 nm reference) with a TECAN photometer (TECAN Spark Control Magellan V2.2, Grödig/Salzburg, Austria). For interassay normalization, three selected positive and three negative controls were used in every plate. OD measurements were ended when the mean value of three selected positive controls reached OD = 1.00 (~1 h). For every breeding period, serum samples of 15 C, 15 FR and 15 FRp chickens were examined in parallel experiments in the same microplate. For the comparison between C and FR chickens, measured OD values in each microplate were divided by the mean OD value of C chickens in the respective microplate. For the comparison between FR and FRp chickens, measured values in each microplate were divided by the mean value of FR chickens in the respective microplate. Therefore, for comparisons of NAb levels between two groups, values are expressed as y = OD(x)/meanOD(Control). For comparisons between the C and FR groups, the C group was set as the control group. For the comparisons between the FR and FRp groups, the FR group was set as the control group.

### 2.3. Statistical Analysis

The d’Agostino–Pearson omnibus normality test was used to assess the normality of the data distribution. For comparison of two independent groups the Mann–Whitney U test was used. The non-parametric Friedman test was used to detect differences between groups.

In addition, principal component analysis (PCA) was used. PCA is a dimensionality reduction method used to identify potential correlations and to reduce the dimensionality of datasets (i.e., the number of variables). The new coordinates of the datasets in the dimensionality-reduced space denote the “scores”, while the “loadings” correspond to the correlations between the original variables and the new dimensions (components). Loading values close to +1 or −1 indicate that a variable strongly influences the component, while estimates close to zero indicate that the variable has a weak or no influence on the component. The input data were centered, and singular value decomposition was applied. Biplots were created to show the scores and loadings in a single plot.

Moreover, to study the correlation among NAb levels, European Production Efficiency Factor (EPEF), genotype and season, we applied two similar multivariate linear regression models with two-way interactions. In the first model, we specified NAb levels as the dependent variable and genotype and season as independent factors, with EPEF as the independent continuous variable. In the second model, we specified EPEF as the dependent variable and genotype and season as independent factors, with NAb levels as independent continuous variables.

In all cases, the significance level was set at 5%, the tests were two-sided and a result was considered significant if the estimated *p*-value (*p*) was less than the significance level. Statistical analysis was performed, and graphs were made in GraphPad Prism version 9.0.0 (GraphPad Software, San Diego, CA, USA).

## 3. Results

### 3.1. Comparison of NAb Levels between Free-Range and Conventional Poultry

Levels of circulating IgM NAbs for the FR (Sasso, *n* = 210) and C (Ross 308, *n* = 210) groups are shown in Figure 1. The FR group exhibited significantly higher levels of IgM NAbs against the four antigens studied. Compared to C group, the FR group exhibited 1.6- and 1.4-fold higher levels for actin and DNA, respectively (*p* < 0.0001). Similarly, for non-self/foreign antigens, the FR group exhibited 1.5- and 2.4-fold higher levels for TNP and LPS, respectively (*p* < 0.0001).

Similarly, levels of circulating IgY NAbs for the C and FR groups are shown in Figure 2. The FR group exhibited significantly higher IgY NAb levels only for LPS, which was estimated to be about 1.2-fold higher when compared to the C group (*p* < 0.0001). A non-significant 1.1-fold difference in levels was observed for actin, while no differences were observed for TNP. Interestingly, levels of NAb against DNA were significantly lower by about 0.6 fold for the FR group in comparison to the C group (*p* < 0.0001).

### 3.2. Association of NAb Levels with Season and Performance

An attempt was also made to assess the impact of seasons on NAb levels and, consequently, on how innate immunity levels are shaped. Levels of circulating IgM and IgY NAbs for the C (Ross 308, *n* = 210) and FR (Sasso, *n* = 210) groups are shown in Figure 3 and Figure 4, respectively. The FR group exhibited significantly higher levels of IgM NAbs against the four antigens studied (*p* < 0.001) for the winter, summer and autumn seasons but not for spring. Moreover, for the C group, IgM NAb levels reached a peak in spring for all antigens, as also observed for IgY NAb levels. On the other hand, levels of IgY NAbs against actin, TNP and LPS had an obvious decline in spring for the FR group, reaching lower levels than those of the C group in the same period. IgY NAb levels against DNA were significantly higher in the C group in all seasons (*p* < 0.0001). Furthermore, DNA and IgY NAb levels exhibited significant differences between the FR and C groups for winter, spring and autumn but not for summer.

In order to further investigate how genotype, season and growth performance (Appendix A) may collectively have an impact on NAb levels, multivariate linear regression analysis was performed using two-way interactions. NAb levels against each of the tested antigens were analyzed separately, specified as the continuous dependent variable in each analysis, while genotype and season were specified as independent factors, with EPEF as an independent continuous variable. Parameter estimates and their significance are shown in Appendix A. Those factors had no influence on levels of IgM NAbs against actin, DNA and TNP; however, interestingly, for LPS, although IgM NAb levels were positively correlated with EPEF (*p* < 0.05), they were negatively correlated with EPEF in summer in the case of slow-growth poultry (*p* < 0.05). On the other hand, levels of IgY NAbs against all tested antigens were positively correlated with EPEF, independent of season and genotype (*p* < 0.05). Specifically, levels of IgY NAbs against TNP were found to be positively correlated with the free-range genotype (*p* < 0.05). Additionally, a positive correlation between anti-LPS IgY NAb levels in summer (*p* < 0.05) and autumn (*p* < 0.01) was found, but with a negative correlation with slow-growth EPEF for the same periods (*p* < 0.05 and *p* < 0.01, respectively).

Subsequently, we reversed the question, asking whether the levels of NAbs against the four antigens collectively, together with genotype and season, may correlate with EPEF. Hence, we performed another multivariate linear regression analysis for each antibody isotype independently and defined the EPEF as the dependent variable and the other parameters as the independent variables. Parameter estimates and significance are shown in Appendix A. In every case, this multivariate model pointed out significant differences in EPEF between fast- and slow-growth poultry (*p* < 0.0001). This model also indicated a positive correlation between EPEF, the fast-growth genotype, season (summer and autumn) and levels of anti-LPS IgM NAbs (*p* < 0.05). IgY NAb levels strongly influenced EPEF, and more specifically, anti-actin NAbs negatively affected EPEF in spring (*p* < 0.05), whereas levels of anti-DNA, anti-TNP and anti-LPS NAbs negatively affected EPEF in autumn (*p* < 0.05).

One of the most critical performance parameters is the mortality rate, which is greatly dependent on the immune status of the poultry. Therefore, we investigated its association with the specificity and levels of NAbs. For that reason, principal component analysis was performed for each antibody isotype independently in order to classify predictor variables according to emergent interrelationship patterns among all NAb specificities and mortality rate. Specifically, five variables were used per PCA analysis (i.e., per NAb isotype and genotype for a total of four PCA analyses): anti-LPS, anti-actin, anti-DNA and anti-TNP NAbs, along with mortality rate. The assumption of independent sampling was met. The assumptions of normality, linear relationships between pairs of variables and the variables being correlated at a moderate level were checked. Two components were rotated based on the criterion of eigenvalues over 1 (Keiser rule) and the scree plot. Appendix A displays the items and component loadings for the rotated components, with loadings less than 0.30 omitted to improve clarity. Figure 5 shows PCA biplots for the first two principal components in each case. The first two principal components explain over 65% of the variability in every case, while the dots and lines refer to the individual scores and loadings, respectively. Results suggest that NAbs are negatively correlated with mortality rate, especially in slow-growth broilers, where this phenomenon is obvious for both IgM and IgY NAbs, while for fast-growth broilers, the same observation applies only to IgY NAbs. The specificities of IgM against LPS and IgY against actin were identified as the most important factors differentiating slow-growing from fast-growing broilers, especially in association with mortality rate.

### 3.3. Comparison of NAb Levels between FR and FRp Groups

Levels of circulating IgM and IgY NAbs for the FRp (Sasso, *n* = 135) and FR (Sasso, *n* = 210) groups are shown in Figure 6 (a and b, respectively). The FRp group exhibited significantly higher levels of IgM NAbs against actin, DNA and TNP but not against LPS. In comparison to the FR group, the FRp group exhibited 1.3- and 1.2-fold higher levels for actin (*p* < 0.0001) and DNA (*p* = 0.0002), respectively. Similarly, for non-self/foreign antigens, the FRp group exhibited 1.3-fold higher levels for TNP (*p* < 0.0001). No significant differences were found between the FR and FRp groups for IgY NAbs against the four tested antigens.

## 4. Discussion

Animal health is an integral part of welfare and a prerequisite for both high performance and quality, as well as safety of animal products for human consumption [2]. However, the modern poultry industry has become more susceptible to several welfare concerns due to intensive selection of broilers. Therefore, in order to improve animal welfare, alternative housing systems are becoming increasingly popular [24].

The influence of different housing systems on behavior, performance and cellular immune parameters has already been reviewed [24,25]. Results diverge, although an increase in heterophils and lymphocytes (H/L) ratio has been correlated with higher stress load [26]. Humoral immune parameters such as antibodies have also been studied, along with behavior and performance, but differences are mainly focused on the concentrations of different immunoglobulin classes in serum or other biological fluids [27]. Therefore, we attempted to utilize specific NAb levels as innate immune system indices that may reflect the health status and performance of broilers raised in different industrial rearing systems.

Innate immunity plays an important role in the survival of organisms, as low levels of innate immunity may be related to enhanced disease susceptibility; conversely, high levels of innate immunity may be related to disease resistance [28,29]. With respect to the humoral counterpart of innate immunity, there is an increasing number of studies reporting the significance of NAbs in health and disease. NAbs have been previously detected in a wide variety of species, e.g., fish, reptiles, domesticated and wild birds and various mammals [30,31,32,33,34]. NAb repertoire and levels may be either shaped through continuous polyclonal stimulation by exogenous cross-reactive pathogens or secreted directly from naturally occurring (auto)reactive B-cell clones, or both [35,36,37]. In general, antibody activity and levels of autoreactive IgM increase along with aging and remain stable later on, resulting in a unique repertoire of conserved auto-/polyspecificities, for every individual [38]. IgM patterns appeared to evolve without any exogenous stimulation, confirming the notion that they are not formed randomly [39]. On the other hand, autoreactive IgG patterns do not expand with age and remain stable at a young age [40], suggesting that IgG profiles could represent ‘antibody fingerprinting’ for each individual [41].

We demonstrated herein that in the final stage of farm breeding of each genotype, levels of IgM NAbs against all tested antigens were significantly higher in slow-growth (Sasso) compared to fast-growth (Ross 308) chickens. Additionally, multivariate regression and principal component analysis showed a negative correlation between IgM NAb levels and mortality in free-range broilers. As the beneficial roles of NAbs have been primarily attributed to secreted IgM, our results indicate the significant supremacy of slow-growth over fast-growth broilers, in line with the notion that resistant genotypes are more efficient in free-range systems, where there is a higher incidence of pathogenic and environmental exposure. Indeed, many efforts have been made by researchers to identify and reproduce resistant genotypes in the poultry industry, as well as in other meat-based industries [42,43]. On the other hand, we are aware of the fact that slow-growth broilers have a longer lifespan with increased exposure to pathogens; therefore, immune activation through innate and specific responses occurs at a higher incidence. Moreover, IgM NAb levels are shaped differently across seasons between these two groups, maintaining higher levels in slow-growth broilers, except in spring, when no significant differences were found. This result probably indicates that during this season, environmental challenges may force a global enhancement in innate immunity (through TLR signaling by pathogen-associated patterns (PAMPs) and danger-associated molecular patterns (DAMPs)), with a more obvious effect on conventional poultry. Since microclimatic conditions in conventional systems are set automatically, the observed enhancement in IgM NAb levels specifically in this season can only be interpreted either as a seasonal local phenomenon or as an adverse event. In contrast to IgM NAb levels, IgY NAbs against actin and TNP exhibited no differences between slow- and fast-growth poultry. Interestingly, levels of IgY NAbs against DNA were found to be significantly higher in fast-growth chickens, probably reflecting increased stress and extended cytotoxicity [14] compared to broilers raised in free-range systems. Moreover, although levels of IgY NAbs against actin and TNP exhibited no differences between slow- and fast-growth poultry, levels of anti-LPS IgY NAbs were higher in slow-growth broilers, possibly indicating increased exposure to pathogens followed by an efficient immune response and, therefore, a higher chance of survival. Additionally, seasonal influence on IgY NAbs showed significantly increased levels in fast-growth broilers compared to slow-growth broilers in spring. This probably indicates that environmental changes from cold to hot periods, such as the transition from winter to spring, could activate more specific immune responses in fast-growth broilers due to a higher stress load than in slow-growth chicks, who could offset these challenges through their enhanced IgM NAb immunity. On the other hand, chickens raised under free-range systems have free access the outdoors at will, and when the weather conditions are unstable and alternating, which is very common between seasons in the area where PINDOS farms are located, they usually stay indoors for safety. That could also explain the increased standard deviation in observed values for anti-LPS IgY NAb levels in slow-growth poultry in spring. Our results with respect to the effect seasonal variation on NAb levels may further be supported by the findings of Hollemans et al., who highlighted the importance of farming conditions and their major impact on NAb formation in the very early stages of chickens’ lives [44].The presence of circulating IgM, IgY and IgA NAbs in chicken sera was previously demonstrated by the research group of H. Parmentier [10,11,12,13,14,15,16,17], although focusing mainly on KLH; other target antigens such as actin, myoglobin, transferrin thyroglobulin, DNA, bovine and human serum albumin have also been tested [38,39,40,41,42,43]. To the best of our knowledge, TNP has never been used as a target of circulating NAbs in chickens. Researchers have focused on the KLH antigen since chickens are not normally exposed to it and, therefore, antibodies against this kind of antigen are considered to be naturally occurring. In our research, we focused on TNP as a target antigen for NAbs because, similarly to KLH, individuals are not normally exposed to it, and the presence of anti-TNP antibodies has been demonstrated in mammals, fish and other vertebrates [22,23]. Interestingly, lower mortality of chickens was associated with higher levels of NAbs binding KLH, establishing the idea that enhanced levels are beneficial for the organism [45,46,47]. Multivariate linear regression analysis also resulted in a positive correlation between the free-range genotype and anti-TNP activity, confirming higher levels of polyreactivity in slow-growth broilers.

In terms of performance, slow-growth broilers exhibited a higher cumulative mortality, lower body weight and higher feed conversion ratio (FCR), which led to a lower European Production Efficiency Factor (EPEF). Far from actual numbers, the higher cumulative mortality in broilers raised in free-range conditions could be attributed to their longevity, as they face a higher incidence of disease prior to environmental exposure outdoors, which usually leads to the loss of more birds than in conventional rearing systems. Our regression analysis confirmed significant changes in levels of IgM and IgY anti-LPS NAbs for slow-growth broilers, especially during spring-to-autumn periods. Interestingly, levels of IgM NAbs against LPS in slow-growth poultry were negatively correlated with EPEF in summer (*p* < 0.05), probably reflecting the higher incidence of bacterial pathogenic infection during the summer period. As mentioned above, IgG (IgY for chickens) autoprofiles could represent an ‘antibody fingerprinting’ of each individual based on immune stimulators to which the individual has been exposed. This is in line with the fact that birds spend more time outdoors when weather conditions are mild; therefore, the time frame of exposure increases. Nevertheless, EPEF was found to be strongly associated with the levels of IgY NAbs against all tested antigens, highlighting the fact that active antibody class switching and/or enhancement in the production of IgY NAbs upon stressors is crucial for the productivity of broilers.

Overall, the increased NAb levels observed in free-range broilers in this study and their association with performance, highlight their potential use as informative biomarkers that could be further utilized in the poultry industry. Similarly to our work, the significance of NAbs in animal welfare and survival has been introduced in other animals in the production sector, such as bovine, cows, calves, pigs and laying hens. [47,48,49,50,51,52,53] As human population grows along with the consumer need for meat products, intensive farming requires stress- and disease-resistant animals for efficient production. To this end, it is important to establish reliable markers to evaluate the effectiveness of different genotypes under alternative farming conditions and our work along with other studies indicate NAbs as possible candidates.

Accordingly, in the poultry industry and, more precisely, in free-range poultry, the particularities of specific management conditions still need to be often re-evaluated, as the high worldwide demand for healthier and safer meat products has been rising steadily for years and is projected to continue [43,44,45,46,47]. New dietary strategies have been proposed based on the use of herbal extracts, which are especially developed to ameliorate the productivity and quality of meat products by enhancing the bird’s immune system, minimizing stress levels, including oxidative stress values, and enhancing immune responsiveness. As a result, numerous beneficial properties have been attributed to such herbal extracts, including antimicrobial, antitumor, antimutagenic, antigenotoxic, analgesic, spasmolytic, anti-inflammatory, antiparasitic, insecticidal and hepatoprotective properties [53,54]. Herein, we studied the potential immune enhancement in free ranged chickens fed a diet containing domestic plant extracts, and our results highlight a slight but significant increase in IgM NAb levels for slow-growth broilers fed with plant extracts when compared to those fed a standard diet. However, there was no difference in mortality rate between the two groups.

## 5. Conclusions

This study was carried out under 3-year industrial-scale production of commercial Ross 308 and Sasso genotype broilers for the evaluation of different rearing systems, with significant conclusions drawn. Overall, serum natural antibodies were proven to be valuable immunotools in the poultry industry, as particularly IgM NAbs against the selected panel antigens are able to differentiate between these two genotypes in terms of their immunological status. Additionally, it was shown that natural immune strengthening could be achieved by dietary supplementation with domestic plant extracts.

## Figures and Tables

**Figure 1 biology-12-00304-f001:**
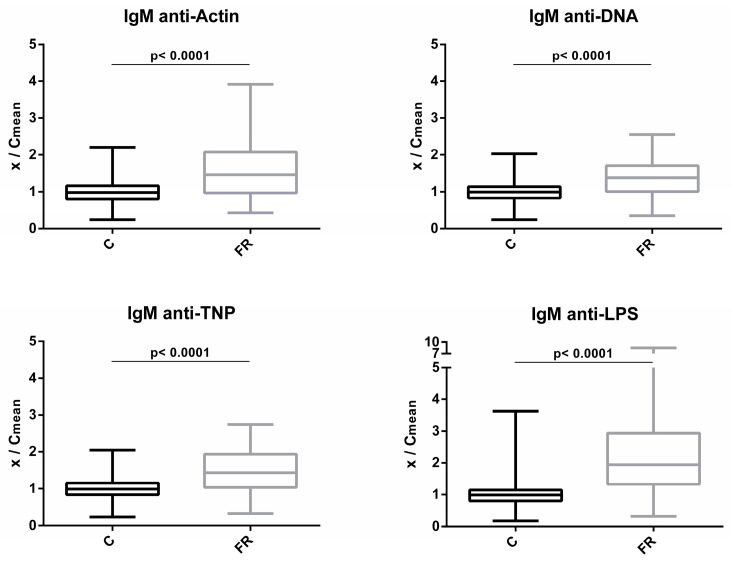
Comparison of IgM NAb levels between fast- and slow-growth broilers: gM NAb levels were estimated by the binding capacity of serum NAbs to actin, DNA, TNP and LPS. On the y axis, values are given as a ratio of individual NAb levels relative to the mean NAb levels of the control group [y = OD(x)/meanOD(C)].

**Figure 2 biology-12-00304-f002:**
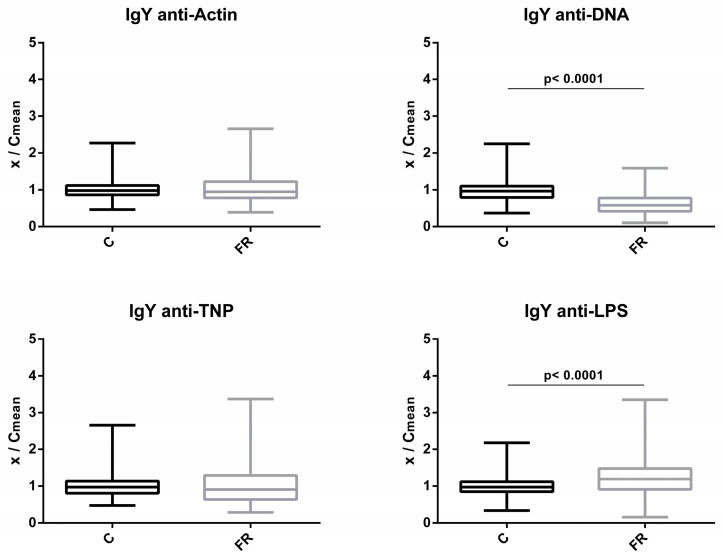
Comparison of IgY NAb levels between fast- and slow-growth broilers: IgY NAb levels were estimated by the binding capacity of serum NAbs to actin, DNA, TNP and LPS. On the y axis, values are given as a ratio of individual NAb levels relative to the mean NAb levels of the control group (y = OD(x)/meanOD(C)).

**Figure 3 biology-12-00304-f003:**
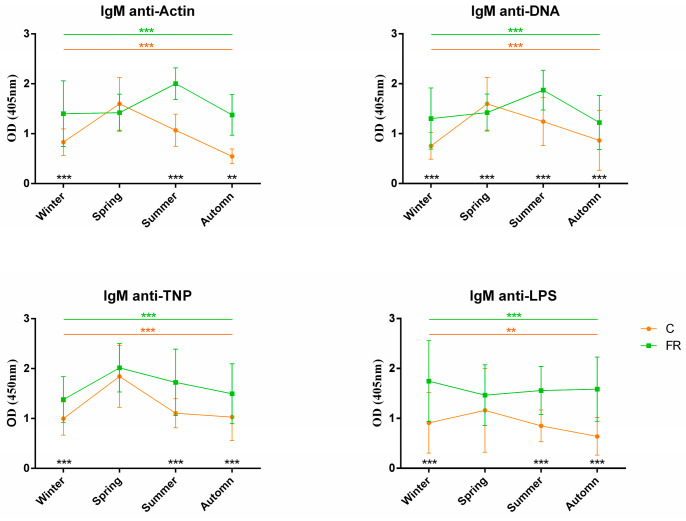
IgM NAb levels for fast- and slow-growth broilers across seasons: IgM NAb levels for the FR (Sasso, *n* = 210) and C (Ross 308, *n* = 210) groups were estimated by the binding capacity of serum NAbs to actin, DNA, TNP and LPS. OD values (mean ± SD) are given on the *y* axis for each year, with season on the x axis. Colored asterisks indicate significant differences among seasons for every genotype, while black asterisks indicate significant differences between the two genotypes (*p* < 0.01 **, *p* < 0.001 ***).

**Figure 4 biology-12-00304-f004:**
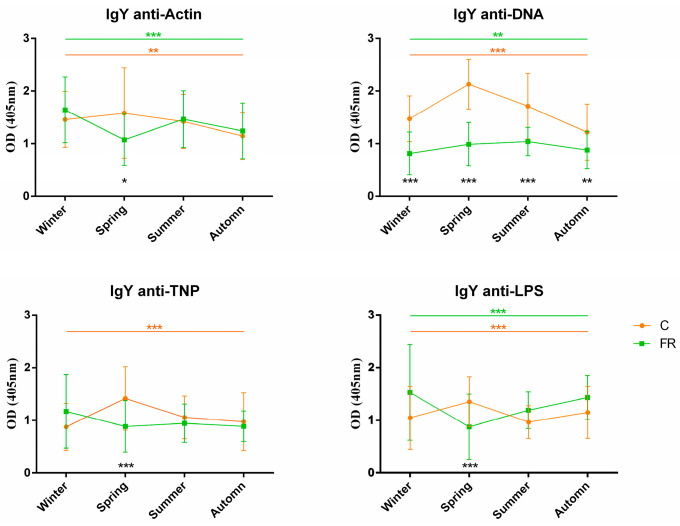
IgY NAb levels for fast- and slow-growth broilers across seasons: IgY NAb levels for the FR (Sasso, *n* = 210) and C (Ross 308, *n* = 210) groups were estimated by the binding capacity of serum NAbs to actin, DNA, TNP and LPS. OD values (mean ± SD) are given on the *y* axis for each year, with season on the *x* axis. Colored asterisks indicate significant differences among seasons for every genotype, while black asterisks indicate significant differences between the two genotypes (*p* < 0.01 **, *p* < 0.001 ***).

**Figure 5 biology-12-00304-f005:**
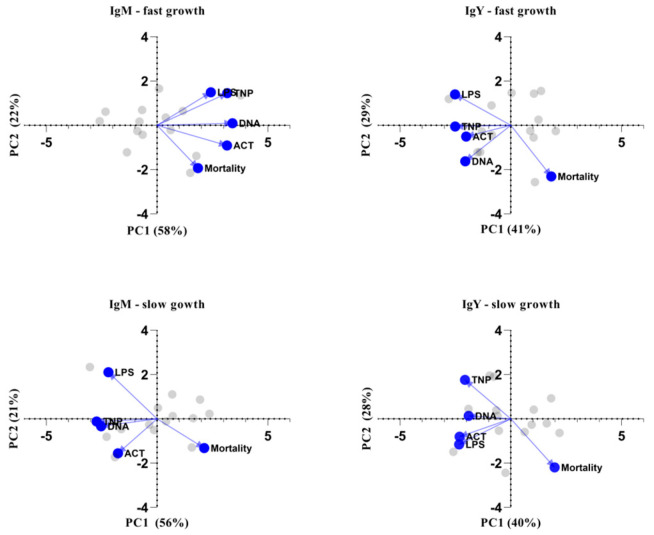
Biplots of principal component analysis. Biplot of the two principal components (PC) showing the individual scores (grey) and the loadings (blue) of the five characteristics: levels of IgM or IgY NAbs against actin, DNA, TNP and LPS, as well as mortality. PCA analysis was performed based eigenvalues meeting the “Kaiser rule”. The variability for every PC is given in the axis title.

**Figure 6 biology-12-00304-f006:**
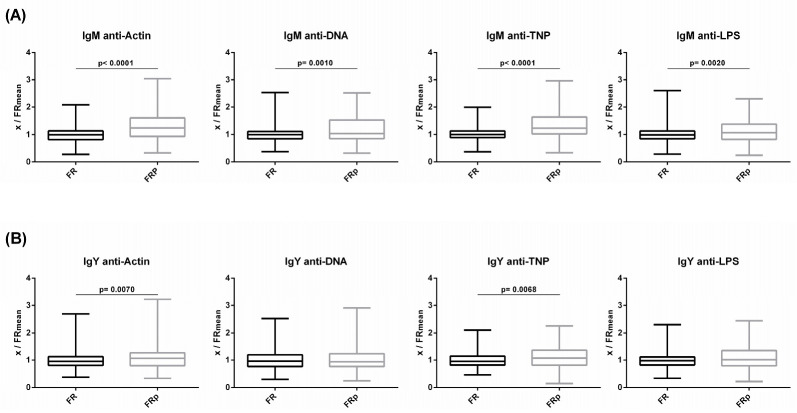
Comparison of NAb levels of slow-growth broilers between those fed a normal diet and those fed a diet supplemented with plant extracts: (**A**) IgM and (**B**) IgY NAb levels were estimated by binding capacity of serum NAbs to actin, DNA, TNP and LPS. On the y axis, values are given as a ratio of individual NAb levels relative to the mean NAb levels of the control group [y = OD(x)/meanOD(FR)].

**Table 1 biology-12-00304-t001:** Dietary formulation of the three tested groups: FR: free-range chickens; FRp: free-range chickens fed with enhanced diet; C: conventional chickens.

Ingredient (kg/ton)	Starter (Days 1–17)			Grower (Days 18–35)			Finisher (Days 36–Slaughter)		
	C	FR	FRp	C	FR	FRp	C	FR	FRp
Corn	200	329	310	150	423	399	0	650	613
Wheat	392	300	283	478	250	236	714	68	64
Soya meal	335	314	296	305	273	257	230	236	223
Phosphoric acid	8.5	6.0	6.0	5.8	6.8	6.4	3.8	7.0	6.0
Limestone	14	14	13	12	13	12	10	11	10
Palm oil	4	0	0	14	10	9	18	17	16
Soya oil	25	17	16	19	9	9	10	0	0
Premix	20.0	19.9	18.8	16.3	15.7	14.8	14.4	11.8	11.1
Oregano premix	-	-	56.6	-	-	56.6	-	-	57.0

**Table 2 biology-12-00304-t002:** Performance analysis of the three tested groups. Values are shown as $\bar{X}$ ± SEM.

	C	FR	FRp
Mortality%	3.81 ± 0.29	4.34 ± 0.52	4.72 ± 0.50
BW	2.59 ± 0.07	2.50 ± 0.06	2.46 ± 0.04
FI	107.00 ± 8.80	89.00 ± 5.36	88.00 ± 5.29
BWG	62.00 ± 3.57 ^a^	36.00 ± 1.81 ^a^	36.00 ± 1.89
FCR	1.74 ± 0.02 ^b^	2.45 ± 0.02 ^b^	2.48 ± 0.20
EPEF	312 ± 4.44 ^b^	145 ± 3.41 ^b^	141 ± 2.86

BW, body weight; FCR, feed conversion ratio; EPEF, European Production Efficiency Factor; FI, feed intake; BWG, body weight Gain; SEM, standard error of means; ^a^ comparison between A and C with significance at *p* ≤ 0.001; ^b^ comparison between A and C with significance at *p* ≤ 0.05.

## Data Availability

The datasets used and/or analyzed during the current study are available from the corresponding author upon reasonable request.

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
