# Peer review of "Levels of Circulating IgM and IgY Natural Antibodies in Broiler Chicks: Association with Genotype and Farming Systems"

_biology, 2023, doi:10.3390/biology12020304_

Round 1

Reviewer 1 Report

The goal of the study by Sarrigeorgiou et al aimed at identifying potential immunological differences between commercial fast- and slow-growth broilers, raised in cage or free-range systems. This manuscript investigates the role of serum natural antibodies in poultry production from multiple perspectives, highlighting naturally-occurring antibodies (NAbs) as potential biomarkers to be exploited in poultry industry. However, these studies were all superficial and the nature of the manuscript is descriptive. Therefore, this novelty of this manuscript was limited. Overall, this seems to be a preliminary work that requires more follow-up experiments in order to be considered for publication.

Comments in detail are listed as following.

1.The manuscript does not accurately reflect the novelty and primary significance of the experiment, which need extensive editing and reformatting in order to be published.

2.As an important participant in innate immunity and acquired immunity, NAbs have many types. However, the reason for the selection of IgM and IgY NAbs for this study was not clearly explained in the background introduction of this manuscript, which needs to be supplemented in the introduction. On the other hand, the last part of the introduction needs to add a summary.

3.The data of the manuscript are unitary, and the innate immunity level of broilers cannot be judged only by the antibody level of the serum. At the same time, the serum antibody level is a dynamic process, and it is recommended to increase the value of IgM and IgY NAbs in each season when describing the association of NAb levels with season and performance.

4.How was this dose the oregano oil as diet supplements administered determined?

5.  The discussion of the manuscript is not compared with the work of other researchers in this or similar fields and lacks reference for researchers in the same field.

Reviewer 2 Report

The paper aims to deepen the knowledge of circulating IgM and IgY NAb levels as potential biomarkers of poultry welfare and productivity in two commercial genotypes in commercial production. The study compares two different situations that refer to the industrial production conditions of both genotypes. although it is important to respect the industrial production conditions, it can sometimes make the research more difficult due to the different conditions. This study is very ambitious and covers a lot of information, which is sometimes a bit confusing. I think the initial description of the experiment and Some points in the discussion should be clarified. I would like to know if this is a retrospective study. If so, it should be indicated.

The research works with two genotypes, Ross 308 raised in cages, and Sasso raised in free range. Were all animals of the same genetics housed on the same farm? this is not specified and should be clarified. If it not, there could be a farm effect.

Authors say that the environmental conditions are similar, should be specified, if all are with natural or artificial photoperiod, number of hours of light/day, humidity, and if the two genotypes were bred in nearby facilities. This should be explained.

The breeding of both genotypes coincides in time, this is important to be able to establish correlations between them. Blood samples are taken at the end of the animals' lives in both genotypes, so the level of immunoglobulins can be affected by many factors.

L116. where was oregano oil and oregano premix obtained from? This should be specified.

L120. Commercial diets were designed for each group (genotype), age and period according to nutritional specifications. The diets for all groups were formulated based on wheat and corn. In this comment, the question arises as to how many diets were used? The authors should say a little more about the feeding program received as it may influence the circulating immunoglobulins.

L127. I understand that in the first batch (19/10/2018-05/12/2018) there were more than 44900 chickens, how were the animals to be sampled chosen?

L128. Since FCR, mortality and performance are discussed later, I think it should be mentioned in material and methods how and when these controls were performed, as well as the control of deaths.

In discussion,

How do higher levels of humoral innate immunity in slow-growing broilers go along with higher mortality? what is the explanation for this?

L306. The paragraph from L306 to L312 seems to be unrelated to the objective of the work, why has this reflection been placed here? I do not understand its purpose

L323. The sentence: "resulting in an 'individual' repertoire of different specificities between subjects" should be better explained.

L330. If levels of autoreactive IgM increase along with aging, in slow-growing breeds where samples were obtained at an older age, it is logical that they present higher levels

L338. How can environmental challenges force a global enhancement of innate immunity? this should be better explained in the manuscript.

L346. However, higher mortality is observed in this genotype. Does this statement not contradict itself?

L348. In free-ranging animals, would they not be more subject to these climatic variations? in Matter and methods, the environmental conditions of each type of production system should be explained, because these conclusions are not clear.

Why has the comparative study of mortality between chickens fed with and without oregano extract not been carried out? It would be a way to ensure one of the conclusions of the work.

Table S1. Why in losses you add 2%.?

In the first two weight data the dot is superfluous

In Figure 3. the statistical differences should be indicated

These points should be addressed before a final decision is made

Reviewer 3 Report

There is little point to be addressed, and the manuscript is almost perfect.

Author Response

We thank the reviewer for the comments. Please find uploaded the revised version of our manuscript. Thank you.

Reviewer 4 Report

Dear authors,

Your manuscript entitled - Circulating IgM and IgY natural antibodies’ levels in broiler chicks: association with genotype and farming systems- is  written very clearly and the paper will be an interesting research addition to the existing literature and research field.

I appreciate your work, you succeded to explain your reserch in a very simple and clear writing, with an interesting content .

Please add some more information requested below:

Simple Summary

Row. 19 . Please rephrase

Within  the European Union, starting from 2012, the conventional cage were banned and only enriched cages with a minimum space of 750 cm2 per /hen or alternative housing systems such as a barn, free-range and organic systems are allowed

Introduction

Row 58-59. Please find another statement regarding poultry rearing worlwide. As I mentioned earlier classic cage system is banned in E.U.

Materials and Methods

2.1 Study design

Regarding fast growth broilers (Ross 308, Aviagen Group, Huntsville, Alabama, USA) raised in conven- tional cage systems used as C, according to ROSS guide,  the  cycle production last 56 days.

Row 118-119. Please add a  few details about microclimate for each system!

Row 121-122. Please add more info about the diet characteristics (ME, CP)!!!!!!

Results and Discussion

Please add a tabel with some results regarding performance parameters: weight, average daily gain, average daily intake, cummulative intake, feed conversion rate

Round 2

Reviewer 1 Report

OK. I have no more questions now. Thank you.

Reviewer 2 Report

The authors have correctly clarified the questions posed to them, although the  line 145 needs to be rewritten .